# Shiga toxin-producing *Escherichia coli* O157:H7 among diarrheic patients and their cattle in Amhara National Regional State, Ethiopia

Tigist Engda[1]*, Belay Tessema[2], Nebiyu Mesifin[3], Anwar Nuru[4], Teshome Belachew[1], Feleke Moges[1]

1 Department of Medical Microbiology, School of Biomedical and Laboratory Sciences, College of Medicine and Health Sciences, University of Gondar, Gondar, Ethiopia, 2 Faculty of Medicine, Institute of Medical Microbiology and Virology, University of Leipzig, Leipzig, Germany, 3 Department of Internal Medicine, College of Medicine and Health Sciences, University of Gondar, Gondar, Ethiopia, 4 College of Veterinary Medicine and Animal Sciences, University of Gondar, Gondar, Ethiopia

* tigiengda@gmail.com

## Abstract

### Background

Shiga toxin-producing *Escherichia coli* O157:H7 (STEC O157:H7) is a zoonotic pathogen that causes diarrhea, hemorrhagic colitis, and hemolytic uremic syndrome worldwide. This study aimed to determine the prevalence, antibiotic susceptibility, and associated risk factors of STEC O157:H7 among diarrheic patients and their cattle.

### Methods

A cross-sectional study was conducted among diarrheic patients and their cattle in Amhara National Regional State, Ethiopia from December- 2020 to June- 2022. A total of 1,149 diarrheic patients and 229 cattle were included in the study. STEC O157:H7 detection was done using culture, latex agglutination test, and polymerase chain reaction on diarrheic stool samples and recto-anal mucosal swabs of cattle. Antibiotic susceptibility tests were performed using disk diffusion techniques. Risk factors association were identified using binary and multivariable logistic regression analysis.

### Results

The overall prevalence of STEC O157:H7 in diarrheic patients and their cattle was 11.1% (128/1149) and 14.4% (33/229) respectively. High percentage of the study subjects were found in under-five children (34.5%). Age less than 5 (AOR: 4.02, 95%CI:1.608–10.058,P = 0.003), and greater than 64 years old (AOR:3.36, 95% CI:1.254–8.986, P = 0.016), presence of diarrheic patient in the house (AOR:2.11, 95%CI:1.309–3.390, P = 0.002), availability of cattle in the house (AOR:2.52, 95%CI:1.261–5.049, P = 0.009), and habit of consuming raw foods (AOR:4.35, 95%CI:2.645–7.148, P = 0.000) were risk factors. Antibiotic resistance was shown in 109(85.2%), and 31(93.9%) isolates from diarrheic patients and their cattle respectively. The highest levels of antibiotic resistance were found to

**Data Availability Statement:** All relevant data, starting from socio-demographic character, prevalence of the isolate bacteria(STEC O157:H7),

associated risk factor of diarrheic patients and their cattle factors are within the manuscript and its Supporting Information files.

**Funding:** There is no specific fund or grant for this study moreover from the authors no one was received specific salary for being a member of this study.

**Competing interests:** The authors have declared that no competing interest exist

tetracycline (54.7%, 69.7%) in diarrheic patients and their cattle respectively. Multiple drug resistance was also observed among 56(43.8%) and 11(33.3%) isolates in diarrheic patients and their cattle respectively.

## Conclusion

Our study showed high prevalence of STEC O157:H7 in diarrheic patients and their cattle. Therefore, health education should be given to the community on how to care for animals, proper sanitation, and the impact of raw food consumption.

## Introduction

Diarrheal disease is a major public health problem responsible for high morbidity and mortality Worldwide. It is among the leading causes of outpatient visits, hospitalization, and the global year of life lost (YLL) in people of all ages [1]. It shares 4% of all deaths and 5% of health losses to disability in the world [2]. It is one of the top ten causes of death, ranking ninth globally, sixth in lower-middle-income countries, and second in low-income countries [3]. Ethiopia ranks fifth with the highest burden of diarrhea and pneumonia in the world [4]. Diarrhea can be caused by different gastrointestinal (GI) pathogens, including viruses, bacteria, and protozoa [5]. Among bacterial pathogens, *Escherichia coli* (*E. coli*) is the most common etiological agent that causes moderate-to-severe diarrhea in low-income countries [6]. This group of *E. coli* is said to be Diarrheagenic *E. coli* (DEC) and is a significant contributor to diarrheal disease throughout the world [7]. Shiga toxin-producing *E. coli* O157:H7 is one of Diarrheagenic *E. coli* [7].

Shiga toxin-producing *E. coli* O157:H7 is typically a food-borne pathogen causing gastroenteritis and bloody diarrhea, and sometimes it leads to hemolytic uremic syndrome (HUS), thrombotic thrombocytopenic purpura (TTP), end-stage renal disease (ESRD), and even to death [8].

It is noted that for the first time, STEC O157:H7 emerged as a human pathogen in the USA in the early 1980s, during large-scale outbreaks of hemorrhagic colitis and HUS [9]. Since then, it has been epidemiologically and clinically important worldwide. Around half of human cases are sporadic, with a seasonal pattern favoring spring and summer. Reports showed that the distribution of STEC O157:H7 differs between regions, potentially influencing the incidence and severity of human disease [10].

Shiga toxin-producing *E. coli* O157:H7 is a zoonotic bacterial pathogen in ruminant animals like cattle. Cattle are the most important reservoir for STEC O157:H7 [11]. However; due to the absence of globotriaosylceramide-3 (Gb3) vascular receptors for Shiga toxin, its colonization to gastrointestinal tracts of cattle is usually asymptomatic [12], and even it cannot be endocytosed and transported to other organs [13]. As a result, it can accumulate in the terminal part of the large intestine and recto-anal junction [14]. Moreover; Shiga toxin-producing *E. coli* O157:H7 can also be found in water, soil, meat, fruit, and vegetable products that are contaminated with ruminants' fecal material [15].

The best treatment of STEC O157:H7 infection is supportive care. Such as; balancing fluid levels and electrolytes and monitoring the possible development of microangiopathic complications such as HUS [8]. Antibiotic therapy is considered to be not beneficial as several antibiotics have been observed to induce the expression and release of Shiga toxins [9].

Even though reports were showing the prevalence of STEC O157:H7 among diarrheic patients in Ethiopia [16–19], still there is limited information in different regions of the country, particularly in the Amhara National Regional State. Therefore, this study aimed to investigate the distribution and antibiotic susceptibility pattern of STEC O157:H7 among diarrheic patients and their cattle in Amhara National Regional State of Ethiopia.

## Materials and methods

### Study area and period

The study was conducted in Comprehensive Specialized Hospitals found in Amhara National Regional States, namely Debre-Markose, Felege-Hiwot, Dessie, Debre-Tabor, and University of Gondar Comprehensive Specialized Hospitals from December 2020 to June 2022. Amhara National Regional State is the second most populous region in the country, Ethiopia. The region has a total population of 20,769,985 of those, 3,492, 000 (16.8%) were urban inhabitants [20].

### Study design

A cross-sectional study design was used to determine the prevalence of STEC O157:H7, its associated factors, and antimicrobial susceptibility patterns among diarrheic patients and their cattle.

### Study population

The source population is all diarrheic patients who had visited the respective Comprehensive Specialized Hospitals for medical services and their cattle. However; These diarrheic patients who are volunteers for the study and fulfill the requirements including giving informed consent and the required amount of stool samples for laboratory analysis and cattle found in the house of STEC O157:H7 positive patients are the study population.

### Inclusion and exclusion criteria

All diarrheic patients who had visited the selective Hospitals for medical services during the study period and cattle found in the house of STEC O157:H7 positive diarrheic patients were included. However, patients who did not gave consent to the study, and patients and cattle who were on antibiotics treatment for the last 7 days of data collection, were excluded from the study.

### Sample size and sampling techniques

The sample size for diarrheic patients was calculated using the statistical formula [6],

$$n = \frac{Z^2 x\, p(1-p)}{d^2}$$

where, n = sample size, P = Prevalence in the target population to have STEC O157:H7 (13.9%) (19), Z = Value of standard normal distribution (Z-statistic) at 95% confidence interval (Z = 1.96), d = desired absolute precision (margins of error) for estimating a single population proportion (d = 2% = 0.02), and the final sample size was 1,149.

A simple random sampling technique was used to select the Hospitals. However, the study participants were selected using systematic random sampling techniques. The sample size was proportionally allocated to the selected Hospitals based on the previous case flow. It was calculated by the formula; $ni = \frac{(Ni \times n)}{N}$, Where ni = sample size allocated for a given site, Ni = total

number of diarrheic patients visited in a given site in the last 12 months of data collection (Debre-Markose = 2776, Felege-Hiwot = 2576, Debre-Tabor = 2489, Dessie = 2489, and University of Gondar = 3970), N = total number of diarrheic patients attended in the respective Hospitals in the last 12 months (14,300) and n = the sample size of the study (1,149). Based on this formula, 223, 207,200, 200, and 319 diarrheic patients were allocated to Debre-Markose, Felege-Hiwot, Debre-Tabor, Dessie, and the University of Gondar compressive specialized Hospitals respectively.

In the case of cattle, all cattle found in STEC O157:H7-positive patients' houses were included in the study and were found to be 229. Other than the wastage of materials, this increases the chance of getting positive cases.

## Data collection tools

**Questionnaire.** A pretested structured questionnaire was used to collect the necessary data. After obtaining written consent and/or assent from each participant, socio-demographic data, clinical information, and information on the availability of cattle and other possible risk factors of diarrheic patients for STEC O157:H7 were collected. The questionnaire was prepared in English using published studies and translated into the local language (Amharic). Once data were collected, responses to each questionnaire were re-translated into English for analysis and report.

## Sample collection and transportation

**Diarrheic stool sample.** After written informed consent and/or assent was obtained, a structured questionnaire was used and completed by laboratory technicians on a face-to-face interview basis. The participants were also asked and guided by the laboratory technicians to give the stool sample. The sample was collected with a clean, detergent-free disposable screw-capped bottle, and labeled with the patient's code number. On-spot gross examination of stool samples was performed to note the type of diarrhea. Then, about 2 gm of each fecal sample was added to culture tubes containing 10ml tryptone soya broth (TSB) for transportation to the laboratory in a cooler box within 2 hours of collection [21].

**Recto anal mucosal swab.** Recto-anal mucosal swab (RAMS) sample is more sensitive than fecal sampling for determining the prevalence of STEC O157:H7 in ruminant animals [22]. The Recto-anal mucosal swab was collected before defecation to minimize fecal contamination [23]. It was used aseptically from each cattle by inserting a sterile cotton-tipped swab about 2-3cm into the anus. Using a circular motion, the entire surface of the recto-anal mucosa was swabbed. Each swab was placed into a 5ml culture tube containing 2ml TSB for transportation to the laboratory in a cooler box within 2 hours of collection.

Both tryptone soya broth-enriched samples were stored in a cold box with ice packs to transport to the University of Gondar Microbiology Laboratory and the samples were processed immediately after arrival, otherwise, they were stored at -20˚C until processed.

**Isolation and characterization of Shiga toxin-producing E. coli O157:H7(S1 File).** The enriched samples were cultured aerobically at 37˚C for 24 hours on MacConkey agar (Oxoid, England). After incubation, isolates' characteristics and reactions on agar media were observed and recorded [24]. Five to ten suspected colonies (pinkish color appearance) were sub-cultured on a separate nutrient agar (Oxoid, England) and confirmed by the biochemical tests.

The Biochemical tests were performed on pink-color colonies to differentiate *E. coli*. The typical biochemical reactions that are considered as *E. coli* are positive for the Indole test, negative on Simon's citrate agar, and urease test, and fermentation of lactose and glucose using

Triple Sugar Iron (TSI) with the production of acid and without hydrogen sulfide ($H_2S$) production [21,25,26].

The identified *E. coli* isolates were sub-cultured on sorbitol MacConkey agar containing cefixime and tellurite (CT-SMAC) (Oxoid, England) media at 37˚C for 24 hours to differentiate Shiga toxin-producing *E. coli* from other *E. coli* strains. Sorbitol fermenters (pinkish colonies) were considered as non-O157:H7 *E. coli* strains whereas the non-sorbitol fermenting isolates (colorless colonies) were supposed to be as presumptively confirmed as Shiga toxin-producing *E. coli* O157:H7 strains [27].

**Serological test.** All non-sorbitol fermenting colonies from the cefixime and tellurite-containing sorbitol-MacConkey agars (CT-SMAC) were serologically confirmed using RIM™ *E. coli* O157:H7 latex test (R24250, Oxoid, Basingstoke, Hants, England). RIM™ *E. coli* O157:H7 latex contains 3 reagents [28]. The particles in each reagent are coated with a different antibody: one against STEC serotype O157, another against STEC serotype H7, and the third with normal rabbit globulin to serve as control latex. A drop of Latex was dispensed into the circle of the reaction card. Using a loop, 10 separate colonies were taken and added into the circle which contained latex reagent. The test latex particles were mixed with fresh colonies of STEC O157:H7, which is positive by CT-SMAC, and in immunochemical reactions, those undergoing agglutination within a minute were registered as positive for STEC O157:H7. The absence of agglutination indicates that the test isolates were not STEC O157:H7. The control Latex reagent identifies non-specific reactions [28].

**Polymerase chain reaction (PCR) analysis.** Deoxyribonucleic acid (DNA) extraction of the isolate was done by boiling and centrifugation. The isolates grown in nutrient broth were harvested, centrifuged at 12,000 rpm for 10 minutes, and the supernatant was discarded. The sediment was then washed with 1.0ml distilled water and vortexed. The isolate sediment was lysed by boiling in a water bath at 95˚C for 10 minutes and the lysate was then centrifuged. Finally, the supernatant was used as a DNA template for PCR [29].

**Amplification of rfb O157 and flic H7 genes.** Isolates were confirmed as STEC O157:H7 by using specific primers for rfb O157 (292 bp) [30,31] and flic H7 genes (625bp) [30,32]. Amplification of rfb O157 and flic H7 are genes encoding for the O157 somatic and H7 flagellar antigens respectively. These were estimated by using adapted primers [31,32] (Table 1). The amplification mixture contains 5µl platinum super II green master mix (Taq DNA polymerase, dNTPs, $MgCl_2$, and reaction buffer at optimal buffer concentration for efficient amplification of DNA templates by PCR, 1.25µl purified DNA, 0.25µl of each forward and reverse primer, then the volume was completed to 3.25µl by deionized water. A negative control was performed by adding 10µl of sterile deionized water and a positive control was performed by adding 1.25µl of known DNA sample (STEC O157:H7; ATCC 43895). All tubes were centrifuged in a microcentrifuge for 10 seconds. Then for the amplification reaction, PCR tubes were transferred to the thermocycler. Polymerase chain reaction conditions consisted of an

**Table 1. Oligonucleotide sequences of the primers used in PCR amplification.**

| Target gene | Primers | Nucleotide sequence (5'-3') | Amplicon size (bp) |
|---|---|---|---|
| **rfbO157** | rfb-F | GTGTCCATTTATACGGACATCCATG | 292 |
|  | rfb-R | CCTATAACGTCATGCCAATATTGCC | 292 |
| **flic H7** | flic-F | GCGCTGTCGAGTTCTATCGAGC | 625 |
|  | flic-R | CAACGGTGACTTTATCGCCATTCC | 625 |

Key: rfb-F = Forward primers, rfb-R = Reverse primers, rfbO157 = rfb O157 gene encoding for the O157 somatic antigens, flicH7 = flic gene encoding H7 flagellar antigen.

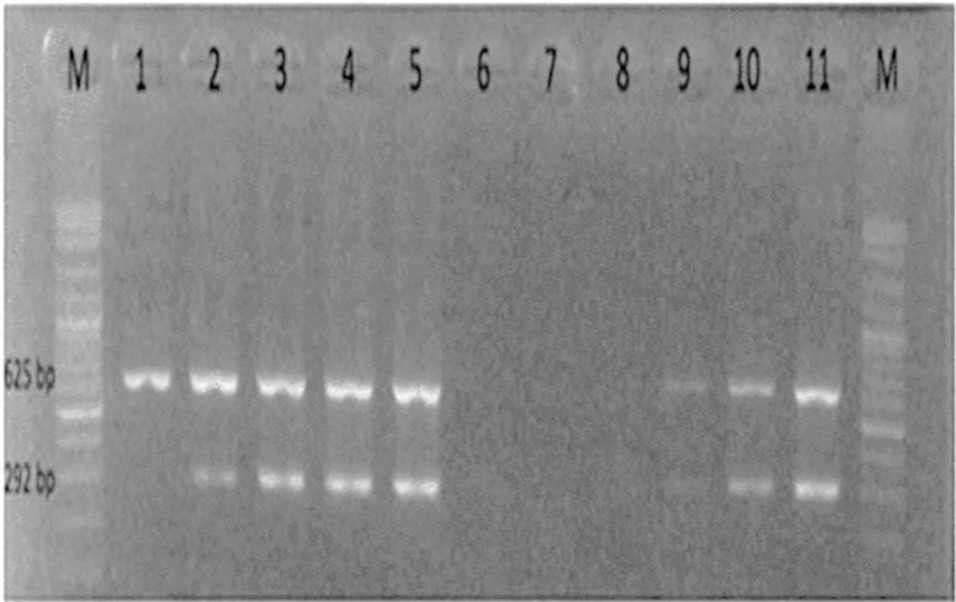

**Fig 1. Agarose Gel Electrophoresis of amplified genes of STEC O157: H7.** M represents the molecular ladder (100bp), lane2 is positive control, and lane6 is negative control. Lane 1,3,4,5,7,8,9,10, &11 represents the samples. Samples on lane 7 and 8 did not show any amplification of the target genes while lanes 1, 3, 4, 5, 9, 10, and 11 showed amplification of rfbO157 = rfb O157 (292bp) gene encoding for the O157 somatic antigens, flicH7 = flic gene (625bp) encoding H7 flagellar antigen as evidenced by the presence of bands. Sample on lane 1 amplified only flicH7 = flic gene (625bp).

initial 98˚C denaturation step for 1 minute followed sequentially by 35 cycles of 98˚C for 10 seconds, 53˚C for 10 seconds, and 72˚C for 45 seconds. The final extension cycle was followed at 72˚C for 5 minutes.

**Agarose gel electrophoresis.** Amplified PCR products were analyzed by gel electrophoresis in 1% agarose gel. The gels were stained with 0.5μl of ethidium bromide (EtBr) per ml and electrophoresed at 120v for 50 minutes using 1xTris-borate-EDTA (TBE) buffer and used a marker DNA ladder of 100bp. The products were visualized with UV illumination and imaged with a gel documentation system (Fig 1).

**Antimicrobial susceptibility test.** Antimicrobial susceptibility testing was done on Mueller-Hinton agar (Oxoid, England) using the disk diffusion technique according to the Kirby-Bauer method [17]. The antimicrobial agents tested were amoxicillin/clavulanate (20/10μg), ceftazidime (30μg), ceftriaxone (30μg), cefixime (5μg), cefuroxime (30μg), sulfamethoxazole/trimethoprim (1.25/23.75μg), ciprofloxacin (5μg), norfloxacin (10μg), tetracycline (30μg), gentamycin (10μg), and chloramphenicol (30μg).

Bacterial inocula were prepared by suspending 4 to 5 freshly grown STEC O157:H7 colonies in 3-5ml sterile physiological saline and turbidity was adjusted to a 0.5 McFarland standard [17]. The sterile cotton swab was dipped, rotated several times, pressed against the wall of the test tube and then swabbed over the entire surface of the Muller-Hinton agar. After the plates were dried, antibiotic-impregnated disks (Oxoid, England) were placed on the surface of the inoculated plates using sterile thumb forceps. The plates were incubated aerobically at 37˚C for 24 hours. Finally, the diameter of the inhibition zone formed around each disk was measured on the black surface using a transparent ruler placed over the plates and recorded. Following CLSI guidelines, the antibiotic susceptibility profile was classified as sensitive (S), intermediate (I), and resistant (R) [35–37].

Moreover; Multiple Drug Resistance, resistance to at least three antibiotic classes, profiles were determined against the commonly used classes of antimicrobials, Cephem class (ceftriaxone, ceftazidime, cefixime, cefuroxime), β-lactam combination class (amoxicillin/clavulanate), Aminoglycoside class (gentamycin), Fluoroquinolone class (ciprofloxacin, norfloxacin), Tetracycline class (tetracycline), Folate pathway inhibitors (sulfamethoxazole/trimethoprim), Phenicols class (chloramphenicol) [33].

## Study variables

**Dependent variables:**—STEC O157:H7 associated diarrhea

**Independent variables**:- are explanatory variables that include socio-demographic behavioral and environmental factors concerning diarrheic patients (age, sex, residence, another diarrheic person in the house, duration of diarrhea, frequency of diarrhea, stool appearance, drinking water source, place of defecations, history of hand washing habit, history of consuming raw food, availability of cattle in the house and sharing of home with cattle), and socio-demographic behavioral factors related to cattle found in the diarrheic patient (age, residence, water source, washing habit of cattle, home of cattle, type of cattle food, purpose of cattle).

## Quality control

The questionnaire was pretested for a week before data collection started. The training was prepared for laboratory technicians to ensure their data collection techniques. Supervision was made at regular times. The questionnaires were checked their completeness and collected at regular times. Appropriate standard strains, *E. coli* (ATCC 25922), and STEC O157:H7 (ATCC 43895) were used as control strains. The test kits of STEC O157:H7 have their quality control material that can be run in parallel with samples, and all test procedures were done strictly following the manufacturer's instructions.

## Data analysis

Data were checked for completeness and entered and analyzed using IBM SPSS Statistics version 25. Descriptive statistics aimed to summarize the study participants' characteristics across the outcome variable were used. The association between the outcome variable and each independent variable was analyzed using bi-variable and multi-variable logistic regression models.

The independent variable which had a significant association with the STEC O157:H7 infection in diarrheic patients at a ≤ 0.2 in the bi-variable logistic regression model was entered into the multivariable logistic regression model to identify the association of risk factors with STEC O157:H7 infection. The assumption of goodness of the model was checked by the Hosmer-Lemeshow test (p = 0.071). The association between the outcome and the independent variable was calculated by using the adjusted odds ratio at a p-value ≤ 0.05 and 95% Confidence Interval.

Data were summarized using frequency tables. For descriptive statistics, standard deviation (SD), frequencies, and percentages were used. In the case of categorical variables, univariable and multivariate analysis with a 95% confidence interval (CI) was performed to measure their association, and a p-value ≤ 0.05 was considered statistically significant.

## Ethical statement

We obtained ethical clearance from the ethical review board of the University of Gondar (V/P/ RCS/051/69). After clearly explaining the aim of the study, the confidentiality of their information, and their full right to refuse or drop participating in the research, all diarrheic patients or

their caregiver signed their consent. Participants' woreda, kebele, specific gote, and phone numbers were registered in the questionnaire. We communicated with STEC O157:H7- positive patients to give information about the disease and to collect cattle samples from those who had cattle in their houses. We tried to reach all cattle found in positive diarrheic patients houses through the developmental agency workers and signed their consent.

## Results

### Socio-demographic characteristics of the diarrheic patients

A total of 1149 diarrheic patients were included in this study. The samples distribution was 441(38.4%) from University of Gondar, 207(18.0%) from Felege-Hiwot, 223(19.4%) from Debre-Markose, 200(17.4%) from Debre-Tabor, and 78(6.8%) from Dessie Comprehensive Specialized Hospitals. Of the total diarrheic patients, 613(53.4%) were male and 396(34.5%) were children under five. A high percentage of the study participants, 719(62.6%) were urban residents (Table 2).

**Table 2. Socio-demographic characteristics of diarrheic patients.**

| Socio-demographic characteristics | Category | Frequency N (%) | STEC O157:H7 | |
|---|---|---|---|---|
| | | | Positive N (%) | Negative N (%) |
| **Age** | <5 years | 396 (34.5) | 56 (4.8) | 340 (29.6) |
| | 5–14 years | 229 (19.9) | 24 (2.1) | 205 (17.8) |
| | 15–24 years | 170 (14.8) | 7 (0.6) | 163 (14.2) |
| | 25–64 years | 142 (12.4) | 11 (1.0) | 131 (11.4) |
| | > 64 years | 212 (18.4) | 30 (2.6) | 182 (15.8) |
| **Sex** | Male | 613 (53.4) | 67 (5.8) | 546 (47.6) |
| | Female | 536 (46.6) | 61(5.3) | 475 (41.3) |
| **Marital status** | Single | 311 (27.1) | 35 (3.0) | 276 (24.0) |
| | Married | 784 (68.2) | 88 (7.7) | 696(60.6) |
| | Divorced | 54 (4.7) | 5 (0.4) | 49 (4.3) |
| **Residence** | Urban | 719 (62.6) | 62 (5.4) | 657 (57.2) |
| | Rural | 430 (37.4) | 66 (5.7) | 364 (31.7) |
| **Educational Status** | Illiterate | 394 (34.3) | 54 (4.7) | 340 (29.6) |
| | Read and write | 259 (22.6) | 29 (2.5) | 230 (20.0) |
| | Primary school completed | 252 (21.9) | 20 (1.7) | 232 (20.2) |
| | Secondary school completed | 123 (10.7) | 16 (1.4) | 107 (9.3) |
| | University graduated | 121 (10.5) | 9 (0.8) | 112 (9.8) |
| **Occupation** | Farmer | 165 (14.4) | 13 (1.1) | 152 (13.2) |
| | Housewife | 288 (25.1) | 44 (3.8) | 244 (21.3) |
| | Merchant | 308 (26.8) | 32 (2.8) | 276 (24.0) |
| | Student | 63 (5.5) | 10 (0.9) | 53 (4.6) |
| | Government Employer | 170 (14.8) | 15 (1.3) | 155 (13.5) |
| | Daily laborer | 92 (8.0) | 9 (0.8) | 83 (7.2) |
| | Private employer | 63 (5.5) | 5 (0.4) | 58 (5.1) |
| **Monthly income** | < 500 birrs | 395 (34.4) | 53 (4.6) | 342 (29.8) |
| | 500–1000 birr | 384 (33.4) | 48 (4.2) | 336 (29.2) |
| | >1000 birr | 370 (32.2) | 27(2.3) | 343 (29.9) |
| **Another diarrheic person in the house** | No | 541(47.1) | 42 (3.6) | 499 (43.5) |
| | Yes | 608 (52.9) | 86 (7.5) | 522 (45.4) |

## Prevalence of Shiga toxin-producing *Escherichia coli* O157:H7

In diarrheic patients, 1149 diarrheic stool samples were collected following the standard procedure. The collected diarrheic stool samples were bloody, mucoid, and watery: 420(36.6%), 390 (33.9%), and 339(29.5%), respectively. Of the total stool samples, 128 stool samples were positive for STEC O157:H7 with rfb O157 and flic H7 genes in PCR with agarose gel Electrophoresis tested (Table 1). As a result, the overall prevalence was 11.1% (95% CI: 0.09–0.13).

The highest percentage of STEC O157:H7 was detected among diarrheic patients who had bloody diarrhea (5.2%), 1 to 3 days duration of diarrhea (5.1%), 4 to 9 times diarrheal frequency (6.7%), patients history for hand washing habit in before and after eating (9.1%), and history of consuming raw food (8.3%) (Table 3).

Only a small proportion of the study participants, 187(16.3%) had a total of 561 cattle in their house (Table 3). However, from 187 diarrheic patients, only 53 were positive for STEC O157:H7 and had 229 cattle. Of these cattle, 123, 47, and 59 were cows, oxen and calves respectively. Samples from these cattle were collected by following standard protocol [23].

**Table 3. Clinical manifestation and behavioral characteristics of diarrheic patients.**

| Clinical manifestation and behavioral characteristics | Category | Frequency N (%) | STEC O157:H7 | |
|---|---|---|---|---|
| | | | Positive N (%) | Negative N (%) |
| **Duration of diarrhea** | 1–3 days | 457 (39.8) | 58 (5.1) | 399 (34.7) |
| | 4–6 days | 421 (36.6) | 43 (3.7) | 378 (32.9) |
| | 7–10 days | 271(23.6) | 27 (2.3) | 244 (21.3) |
| **Frequency of diarrhea** | ≤ 3 | 724 (63.0) | 41 (3.6) | 683 (59.4) |
| | 4–9 | 384(33.4) | 77 (6.7) | 307(26.7) |
| | > 9 | 41(3.6) | 10 (0.9) | 31(2.8) |
| **Stool appearance** | Watery | 339 (29.5) | 20 (1.7) | 319 (27.8) |
| | Mucoid | 390 (33.9) | 48(4.2) | 342(29.8) |
| | Bloody | 420 (36.6) | 60 (5.2) | 360 (31.3) |
| **Drinking water source** | Tap | 708(61.6) | 59 (5.1) | 649 (56.5) |
| | Water well | 404(35.2) | 64 (5.6) | 340 (29.6) |
| | Surface | 37(3.2) | 5 (0.4) | 32 (2.8) |
| **Diarrheic patient water treatment** | No | 456 (39.7) | 75 (6.5) | 381(33.2) |
| | Filtering | 391(34.0) | 27 (2.3) | 364 (31.7) |
| | Aqua tab or Water guard | 161(14.0) | 19(1.6) | 143(12.4) |
| | Boiling | 141(12.3) | 8 (0.7) | 133 (11.6) |
| **Place of defecations** | Modern toilet | 217(18.9) | 16 (1.4) | 201(17.5) |
| | Traditional toilet | 583(50.7) | 51 (4.4) | 532(46.3) |
| | Open field | 160(13.9) | 30 (2.6) | 130 (11.3) |
| | Public toilet | 189(16.5) | 31 (2.7) | 158 (13.8) |
| **History of hand washing habit** | Before eating and after the toilet | 249 (21.7) | 13 (1.1) | 236 (20.5) |
| | Before preparing food and washing house utilities | 168 (14.6) | 10 (0.9) | 158(13.8) |
| | Before and after eating | 732 (63.7) | 105 (9.1) | 627 (54.6) |
| **History of consuming raw food** | Yes | 604 (52.6) | 96 (8.3) | 508 (44.2) |
| | No | 545 (47.4) | 32 (2.8) | 513 (44.6) |
| **Availability of cattle in the house** | Yes | 187 (16.3) | 53 (4.6) | 134 (11.) |
| | No | 962 (83.7) | 74 (6.4) | 888 (77.3) |
| **Sharing of home with cattle** | Yes | 103 (9.0) | 33 (2.9) | 70(6.1) |
| | No | 1046 (91.0) | 95 (8.2) | 951(82.8) |

The recto mucosal swab samples of these 33 cattle, (18 cows, 6 oxen and 9 calves) were PCR positive for STEC O157:H7 with rfb O157 and flic H7 genes (Table 1). As a result, the overall prevalence of STEC O157:H7 in cattle was 14.4% (95%CI:0.39–0.52).

## Associated risk factors for Shiga toxin-producing *Escherichia coli* O157:H7

In this study, different variables were considered during the bivariate analysis. These variables were taken as a possible risk factor for STEC O157:H7 infections. Comparatively a higher prevalence of STEC O157:H7 was found among under 5 children and over 64 years of age. Moreover; patients who were rural residents, had mucoid and bloody diarrhea, had low monthly income, were poor in regular hand washing activities before and after eating, had the habit of consuming raw food, and had cattle were risk factors for STEC O157:H7.

In multivariable analysis, diarrheic patients of under-five children were 4(AOR = 4.02, 95% CI; 1.608–10.058, P = 0.003) times and old ages greater than 64 years were 3(AOR = 3.357, 95% CI: 1.254–8.986, P = 0.016) times more likely to have STEC O157:H7 associated diarrhea than other age groups (Table 4).

Diarrheic patients who had an experience of consuming raw foods were 4(AOR = 4.35, 95% CI: 2.645–7.148, p = 0.000) times more likely to have STEC O157:H7 associated diarrhea than their counterparts. Those diarrheic patients who had cattle in their house were 3 (AOR = 2.52 95% CI: 1.261–5.049, P = 0.009) times more likely to have STEC O157:H7 associated diarrhea than patients who did not have (Table 4).

## Antimicrobial susceptibility profile of Shiga toxin-producing *E. coli* O157: H7

All STEC O157:H7 isolates in diarrheic patients and their cattle were subjected to an antimicrobial susceptibility test with 11 commonly prescribed antimicrobial drugs. Antibiotic resistances were shown in 109(85.2%) and 31(93.9%) STEC O157:H7 isolates from diarrheic patient and their cattle respectively.

Shiga toxin-producing *E. coli* O157:H7 isolates exhibited high level of antibiotic resistance to tetracycline 70(54.7%), amoxicillin/clavulanate 68(53.1%), and sulfamethoxazole/ trimethoprim 56(43.8%) in diarrheic patients, and tetracycline 23(69.7%), Amoxicillin/ clavulanate, 15 (45.4%), Sulfamethoxazole/trimethoprim 16(48.5%) in cattle. However; low level of antibiotic resistance was found in chloramphenicol 12(9.4%), norfloxacin 12(9.4%), and ciprofloxacin 16 (12.5%) in diarrheic patients, and chloramphenicol 1(3.03%), norfloxacin 2(6.1%) and ciprofloxacin 4(12.1%) in cattle (Table 5).

In isolates from diarrheic patients, only 11.7% were not resistant to any of the selected antibiotics. However; 43.8% showed Multiple Drug Resistance, resistance to at least three antibiotic classes. From these 3.1% and 0.8% of the isolates were resistant to six and seven antibiotic classes respectively (Table 6).

In cattle isolates, only 6.1% were not resistant to any of the selected antibiotics. However; 33.3% of the isolates showed Multiple Drug Resistance. From these 6.1% and 3.0% of the isolates were resistant to five and six antibiotic classes respectively (Table 7).

## Discussion

STEC O157:H7 is one cause of diarrheal disease and is mostly transmitted by cattle. Cattle are the main natural reservoirs for it [34]. This study was conducted to evaluate the prevalence and antimicrobial profiles of STEC O157:H7 isolates in diarrheic patient and their cattle. Our findings showed that among 1149 diarrheic stool samples, 128 were positive for the organism. The highest prevalence was detected among under- five children. The finding also pointed out

**Table 4. Associated risk factors of Shiga toxin-producing *E. coli* O157:H7 among diarrheic patients.**

| Variables | Category | COR (95% CI) | P-value | AOR (95% CI) | P-value |
|---|---|---|---|---|---|
| **Age** | <5 years | 3.835(1.710–8.601) | 0.001 | 4.02(1.608–10.058) | 0.003 |
| | 5–14 years | 2.726(1.146–6.486) | 0.023 | 2.39(0.905–6.337) | 0.079 |
| | 15–24 years | 1.00 | | 1.00 | |
| | 25–64 years | 1.955(0.737–5.185) | 0.178 | 1.63 (0.546–4.867) | 0.381 |
| | > 64 years | 3.838(1.642–8.975) | 0.002 | 3.36 (1.254–8.986) | 0.016 |
| **Residence** | Urban | 1.00 | | 1.00 | |
| | Rural | 1.921(1.328–2.780) | 0.001 | | |
| **Monthly income** | <500 birr | 1.969(1.210–3.204) | 0.006 | 1.55 (0.842–2.846) | 0.160 |
| | 500–1000 birr | 1.815(1.106–2.977) | 0.018 | 2.39 (1.285–4.456) | 0.006 |
| | >1000 birr | 1.00 | | 1.00 | |
| **Another diarrheic patient in the house** | No | 1.00 | | 1.00 | |
| | Yes | 1.957(1.327–2.888) | 0.001 | 2.11(1.309–3.390) | 0.002 |
| **Frequency of diarrhea/ day** | ≤ 3 | 1.00 | | 1.00 | |
| | 4–9 | 4.178(2.795–6.247) | 0.000 | 4.34 (2.730–6.902) | 0.000 |
| | > 9 | 5.374(2.465–11.715) | 0.000 | 4.23 (1.669–10.724) | 0.002 |
| **Drinking water source** | Tap | 1.00 | | | |
| | Water well | 2.071(1.420–3.020) | 0.000 | 1.294(0.643–2.604) | 0.470 |
| | Surface | 1.719(0.645–4.577) | 0.278 | 1.983(0.543–7.247) | 0.300 |
| **Drinking water treatment** | No | 3.273(1.538–6.964) | 0.002 | 2.228(0.704–7.048) | 0.173 |
| | Filtering | 1.233(0.547–2.762) | 0.614 | 1.173(0.426–3.228) | 0.757 |
| | Water guard | 2.093(0.881–4.973) | 0.095 | 2.945(0.985–8.805) | 0.053 |
| | Boiling | 1.00 | | | |
| **Place of defecation** | Modern | 1.00 | | | |
| | Traditional Latrine | 1.204(0.671–2.161) | 0.533 | | |
| | Open field | 2.899(1.520–5.529) | 0.001 | | |
| | Public latrine | 2.465(1.302–4.667) | 0.006 | | |
| **History of hand washing habit** | Before eating and after the toilet | 1.00 | | 1.00 | |
| | Before preparing food and washing house utilities | 1.149(0.492–2.685) | 0.748 | 0.783(0.296–2.073) | 0.623 |
| | After eating | 3.040(1.676–5.514) | 0.000 | 3.55 (1.766–7.143) | 0.000 |
| **History of consuming raw food** | Yes | 3.030(1.994–4.604) | 0.000 | 4.35 (2.645–7.148) | 0.000 |
| | No | 1.00 | | 1.00 | |
| **Availability of cattle in the house** | Yes | 4.872(3.281–7.234) | 0.000 | 2.52(1.261–5.049) | 0.009 |
| | No | 1.00 | | 1.00 | |
| **Sharing of home with cattle** | Yes | 4.719(2.966–7.500) | 0.000 | 2.03(0.892–4.614) | 0.092 |
| | No | 1.00 | | | |

that 11.7% of isolates from diarrheic patients and 6.1% of isolates from cattle were not resistant to any of the selected antibiotics. However; Multiple Drug Resistance from the selected antibiotic classes was found in 43.8% of diarrheic patients and 33.3% of cattle.

The overall prevalence of STEC O157:H7 in this study for diarrheic patients was 11.1%. This was in agreement with reports from Bahir Dar, Ethiopia (13.9%) [18], and Western Kenya (11.1%) [35]. However, our result was lower than those reported from Eastern Ethiopia (15.3%) [16], Eastern Cape Town, South Africa (17%) [36], and Maasai land Kenya (24.1%) [37]. On the contrary, this finding is higher than the studies reported from Gondar (1.9%) [38], Sebeta town, Ethiopia (3.2%) [13], Bishoftu town, Ethiopia (2.9%) [39], Debre Berhan cities, Ethiopia (0%) [19], Nairobi Kenya (0.2%) [40], Southern Mozambique (1.9%) [41], Morogoro Tanzania (3.6%) [42], Burkina Faso (9.67%) [43], Chattogram, Bangladesh (1.45%) [44],

**Table 5. Antibiotic susceptibility profiles for STEC O157:H7 isolates from diarrheic patients and their cattle.**

| Antibiotic | STEC O157:H7 from diarrheic Patients (N = 128) | | | STEC O157:H7 from Cattle (N = 33) | | |
|---|---|---|---|---|---|---|
| | S N (%) | I N (%) | R N (%) | S N (%) | I N (%) | R N (%) |
| Amoxicillin/ Clavulanate | 48 (37.5) | 12 (9.4) | 68 (53.1) | 12(36.4) | 6 (18.2) | 15 (45.4) |
| Ceftazidime | 106 (82.8) | 4 (3.1) | 18 (14.1) | 25 (75.7) | 2 (6.1) | 6 (18.2) |
| Ceftriaxone | 99 (77.3) | 12 (9.4) | 17 (13.3) | 18 (54.5) | 9 (27.3) | 6 (18.2) |
| Cefixime | 101 (78.9) | 8 (6.3) | 19 (14.9) | 17 (51.5) | 10 (30.3) | 6 (18.2) |
| Cefuroxime | 92 (71.9) | 2 (1.6) | 34 (26.5) | 22 (66.7) | 2 (6.0) | 9 (27.3) |
| Sulfamethoxazole/ Trimethoprim | 67 (52.3) | 5 (3.9) | 56 (43.8) | 15 (45.4) | 2(6.1) | 16 (48.5) |
| Ciprofloxacin | 103 (80.5) | 9 (7.0) | 16 (12.5) | 24(72.7) | 5 (15.2) | 4 (12.1) |
| Norfloxacin | 112 (87.5) | 4 (3.1) | 12 (9.4) | 28(84.9) | 3 (9.0) | 2 (6.1) |
| Tetracycline | 51 (39.8) | 7 (5.5) | 70 (54.7) | 9 (27.3) | 1(3.0) | 23 (69.7) |
| Gentamycin | 63 (49.2) | 10 (7.8) | 55 (43) | 10 (30.3) | 17 (5.5) | 6 (18.2) |
| Chloramphenicol | 109 (85.1) | 7 (5.5) | 12 (9.4) | 28 (84.9) | 4(12.1) | 1 (3.0) |

Key: AMC: Amoxicillin/Clavulanate, CAZ: Ceftazidime, CTR: Ceftriaxone, CFM: Cefixime, CXM: Cefuroxime, SXT: CIP: Ciprofloxacin, NX: Norfloxacin, TC: Tetracycline, CN: Gentamycin, CHL: Chloramphenicol. MDR = Multi—Drug Resistance, S = Susceptible, I = Intermediate, and R = Resistance.

Sudan (5%) [45], plateau state, Nigeria (5%) [46], and France (3%) [47]. The relatively a higher prevalence of the organism found in the current finding might be because of a higher level of traditional breeding practice of animals and animal -to- human interaction in the study subjects.

In this study, STEC O157:H7 was isolated in all age groups. However, the isolation rate was high in age groups of less than 5 years i.e., 5.1% STEC O157:H7 positive from the total study population. This finding was lower than studies reported from Bahir Dar (13.9%) [18], Eastern Ethiopia (15.3%) [16], Western Kenya (11.1%) [35], and Eastern Cape, South Africa (17%) [36]. Even if, the result is lower than in other studies because of study population variation. In our study, under-five children were risk group for the infection. It is due to immature immune systems and poor in hygiene practices among children and their care givers at the time of food preparation, handling, transportation and feeding.

Hand washing habit was identified as a predisposing factor for O157:H7 infection as evidenced by multivariate analysis. Patients who had an experience of hand washing after eating were about 4 times more likely to have STEC O157:H7 infections. This finding is in line with the study reported from Bahir Dar town [18].

Consuming raw vegetables, fruits, and undercooked foods is 4 times significantly associated with the prevalence of STEC O157:H7. This finding is higher than the study done in Eastern Ethiopia [16] and Bishoftu town, Ethiopia [39]. This may be due to a high level of animal manure contamination of vegetables through untreated surface water or animal manure usage as a fertilizer.

It was found that diarrheic patients who had low monthly income were 2 times more likely to have STEC O157:H7 than those who had higher monthly income. This is because income is a key instrument in applying a preventive mechanism for the predisposing factor of STEC O157:H7 infection.

The prevalence of STEC O157:H7 in cattle was 14.4%. This is higher than the study done in Debre-Birhan, Ethiopia (0.81%) [19], Hawassa, Ethiopia (4.7%) [34], Jimma, Ethiopia (7.3%) [48], Addis Ababa Municipal Abattoir, Ethiopia (6.4%) [49], Debre-Zeit, Ethiopia (7%) [50],

**Table 6. Multiple antibiotic resistance patterns of STEC O157:H7 isolates from diarrheic patients.**

| Antibiotic Disk | Total Isolates N (%) | Antibiogram |
|---|---|---|
| Isolates sensitive to all antibiotics (19) * | 19 (11.7) | R0 |
| TC (5), SXT (6), AMC (3), CXM (3), CFM (1), CN (2) | 20 (15.6) | R1 |
| AMC & TC (5), SXT & CIP (1), SXT & TC (10), AMC & CN (1), AMC& SXT (3), AMC & CXM (2), CXM & TC (4), CIP & CN (1), TC, CAZ & CTR (1), AMC, CTR & CXM (1) CIP, NX & TC (1), AMC, CTR & CFM (3) | 33 (25.8) | R2 |
| SXT, TC & CN(1), AMC, SXT, & CIP(1), AMC, SXT & CAF(1), AMC, CXM & TC(2), SXT, TC & CN(2), AMC, CAZ & TC (1), AMC, SXT & TC(7), AMC, TC & CHL(2), AMC, CXM & CN(3), AMC, TC & CN(1), AMC, SXT & CN(1), SXT, CIP, NX & TC(1), AMC, CFM, CXM & TC(1), AMC, CAZ & CTR, CFM & CN(1), AMC, CAZ, CTR, CFM & TC(1), | 26 (20.3) | R3 |
| AMC, CTR, SXT & TC(2), AMC, TC, CN & CHL(1), AMC, SXT, TC & CN(1), AMC, CXM, SXT & TC(1), AMC, CN, CHL, CAZ & CXM(2), AMC, CAZ, CXM, TC & CN(1), AMC, SXT, CIP, NX & TC(1), AMC, CAZ, CTR, CFM, SXT & TC(1), AMC, CAZ, CTR, CFM, TC & CN(1), AMC, CAZ, CFM, CXM, TC & CN(1), AMC, CTR, CFM, CXM, SXT & TC (1), CAZ, CFM, CXM, SXT, CIP, NX & TC(1), AMC, CAZ, CTR, CFM, CXM, SXT & TC(2), AMC, CAZ, CTR, CFM, CXM, SXT, CIP & NX(1) | 17 (13.3) | R4 |
| AMC, CXM, SXT, TC & CHL (1), AMC, CXM, SXT, TC & CN(1), AMC, SXT, TC, CN & CHL(1), AMC, SXT, CIP, NX, TC & CN(1), AMC, CAZ, CXM, CIP, TC & CHL(1), AMC, CAZ, CXM, SXT, TC & CN(1), AMC, CAZ, CTR, CFM, SXT, CIP, NX & TC(1), AMC, CAZ, CTR, CFM, CXM, SXT, CIP, NX & TC(1) | 8 (6.3) | R5 |
| AMC, CXM, SXT, TC, CN & CHL (1), AMC, CFM, SXT, CIP, NX, CN & CHL (1), AMC, CXM, SXT, CIP, NX, TC & CN (2) | 4 (3.1) | R6 |
| AMC, CTR, CFM, SXT, CIP, NX, TC, CN & CHL (1) | 1 (0.8) | R7 |
| Total Non-MDR | 72 (56.3) | |
| Total MDR | 56 (43.8) | |
| Total isolates | 128 (100) | |

**Key**: R1: Resistance to one antibiotic class; R2: Resistance to two antibiotic classes; R3: Resistance to three antibiotic classes; R4: Resistance to four antibiotic classes; R5: Resistance to five antibiotic classes; R6: Resistance to six antibiotic classes; R7: Resistance to more than six antibiotic classes, *19 isolates had a characteristic of Sensitive and Intermediate for all the selected antibiotics. Number in bracket: the number of isolates that showed resistance to the listed antibiotic classes.

Sudan (8%) [45], and Nigeria (0.2%) [51]. However, it is lower than the study reported in Iraqi (91.25%) [52], and Mexico (22.03%) [39]. These variations among different studies could be attributed to differences in geographical area, and the type of samples taken since recto-anal mucosal swab sample is more sensitive than fecal sampling for determining the prevalence of STEC O157:H7 in ruminant animals [6].

The availability of cattle in the diarrheic patient house was about 3 times more likely to have STEC O157:H7 compared to the counterpart. This finding is in line with the report in Eastern Ethiopia [16], and Harare, Zimbabwe [53]. However, it was lower than the report from Germany [54]. The ruminant, especially cattle had been identified as the major reservoir of STEC O157:H7, there could be cross-infection to the diarrheic patients through either direct live animal contact or with animal manure. Animal manure can contaminate food, water, and the environment, and some studies reported that contact with cattle and living in or visiting a place with farm animals are risk factors for the incidence of STEC O157:H7 infection [55].

In this study, STEC O157:H7 isolates (n = 128) in diarrheic patients showed antibiotic resistance at varying degrees. Of all the antimicrobials tested, the highest resistance was found in

**Table 7. Multiple antibiotic resistance patterns of STEC O157:H7 isolates from cattle.**

| Antibiotic Disk | Total Isolates N (%) | Antibiogram |
|---|---|---|
| Isolates sensitive to all antibiotics (2) * | 2 (6.1) | R0 |
| SXT (1), TC (3), CXM (1), CFM (1), AMC (1) | 7 (21.2) | R1 |
| AMC &TC (1), SXT & CIP (2), SXT & TC (7), TC, CAZ & CTR (1), AMC, CTR & CXM (1), AMC, CTR & CFM (1) | 13 (39.4) | R2 |
| AMC, TC & CN (1), AMC, CXM & TC (2) | 3 (9.1) | R3 |
| AMC, TC, CN & CHL (1), AMC, SXT, TC & CN (1), AMC, CAZ, CTR, CFM, SXT & TC (1), AMC, CAZ, CFM, CXM, TC & CN (1), AMC, CAZ, CTR, CFM, CXM, SXT & TC (1) | 5 (15.2) | R4 |
| AMC, CAZ, CTR, CFM, CXM, SXT, CIP, NX & TC (1), AMC, CAZ, CXM, SXT, TC & CN (1) | 2 (6.1) | R5 |
| AMC, CXM, SXT, CIP, NX, TC & CN (1) | 1 (3.0) | R6 |
| Total Non-MDR | 22 (66.7) | |
| Total MDR | 11 (33.3) | |
| Total isolates | 33 (100) | |

**Key**: R1: Resistance to one antibiotic class; R2: Resistance to two antibiotic classes; R3: Resistance to three antibiotic classes; R4: Resistance to four antibiotic classes; R5: Resistance to five antibiotic classes; R6: Resistance to six antibiotic classes; R7: Resistance to seven antibiotic classes. * 2 isolates had a characteristic of Sensitive and Intermediate for all the selected antibiotics. Number in bracket: the number of isolates that showed resistance to the listed antibiotic classes.

tetracycline 70(54.7%), and Amoxicillin/Clavulanate 68(53,1%). Among the STEC O157:H7 isolates from cattle RAMS, the highest resistances were found in tetracycline 23(69.7%), Sulfamethoxazole/Trimethoprim 16(48.5%), and Amoxicillin/clavulanate 15(45.4%). This finding is in agreement with the report in Bahir Dar [18] but is different from Cape Town [50]. This difference may be due to differences in the practice of prescribing the selected antibiotic between the two countries and even the practice of misuse of antibiotics.

Even if, the place of defecation, level of contamination of the water source, and sharing of home with cattle were not found associated with the occurrence of STEC O157:H7. However, other researchers reported open field defecation, drinking contaminated water sources, the presence of the diarrheic person in the house, and the use of common house with ruminants including cattle as a predictor for the occurrence of STEC O157:H7 [16,56,57]. The detection of STEC O157:H7 in human and their cattle calls for further epidemiological assessment to detect the source of infection, and means of transmission, and follow a prevention system.

As to the limitations of this study, we have focused only on STEC O157:H7 strain, but other pathogenic non-STEC O157:H7 serogroups causing illnesses in human needs to be identified.

## Conclusion

The present study revealed a high occurrence of STEC O157:H7 in diarrheic patients and their cattle. Age, patients with bloody diarrhea, the presence of cattle in their house, the presence of another diarrheic person in the house, and the habit of consuming raw foods were more important risk factors for STEC O157:H7 infections than their counterparts. The study also pointed out the presence of resistant STEC O157:H7 to one or more antimicrobials that may lead to a potential public threat. Generally, this study could give an insight into the prevalence of STEC O157:H7 and its public health significance associated with contact cattle, consumption of raw food, and poor personal hygienic practices. Therefore, special attention and awareness should be given to teaching the community on how to give care to ruminant animals and

their home, proper sanitation measures for raw foods, and hand washing habits before preparing food, before eating, and after giving care to patients, and animals. Moreover, medical professionals and veterinarians should work together to give health education to the public concerning STEC O157:H7.

## Supporting information

**S1 File. Laboratory Standard Operational Procedure (SOP) (media used, laboratory procedures, and antimicrobial disks used for their interpretation).**
(DOCX)

## Acknowledgments

We would like to thank the University of Gondar for reviewed our study proposal and awarded ethical clearance for the study. All of the study participants, data collectors, and other staff at the respective hospitals are acknowledged for their cooperation during the data and sample collection processes.

## Author Contributions

**Conceptualization:** Tigist Engda, Belay Tessema, Feleke Moges.

**Data curation:** Tigist Engda, Belay Tessema, Nebiyu Mesifin, Anwar Nuru, Teshome Belachew, Feleke Moges.

**Formal analysis:** Tigist Engda, Feleke Moges.

**Investigation:** Tigist Engda.

**Methodology:** Tigist Engda, Belay Tessema, Nebiyu Mesifin, Anwar Nuru, Teshome Belachew, Feleke Moges.

**Project administration:** Tigist Engda.

**Resources:** Tigist Engda.

**Software:** Tigist Engda, Belay Tessema, Nebiyu Mesifin, Feleke Moges.

**Supervision:** Tigist Engda.

**Validation:** Tigist Engda, Belay Tessema, Nebiyu Mesifin, Anwar Nuru, Teshome Belachew, Feleke Moges.

**Visualization:** Tigist Engda, Feleke Moges.

**Writing – original draft:** Tigist Engda.

**Writing – review & editing:** Tigist Engda, Belay Tessema, Nebiyu Mesifin, Anwar Nuru, Feleke Moges.

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
