## [Decision Letter · Decision Letter 0]

14 Jul 2023

PONE-D-23-13678Shiga toxin-producing Escherichia coli O157:H7 among diarrheic patients and their cattle in Amhara National Regional State, EthiopiaPLOS ONE

Dear Dr Tigist Engda

Thank you for submitting your manuscript to PLOS ONE. After careful consideration, we feel that it has merit but does not fully meet PLOS ONE’s publication criteria as it currently stands. Therefore, we invite you to submit a revised version of the manuscript that addresses the points raised during the review process.

Please include the following items when submitting your revised manuscript:A rebuttal letter that responds to each point raised by the academic editor and reviewer(s). You should upload this letter as a separate file labeled 'Response to Reviewers'.A marked-up copy of your manuscript that highlights changes made to the original version. You should upload this as a separate file labeled 'Revised Manuscript with Track Changes'.An unmarked version of your revised paper without tracked changes. You should upload this as a separate file labeled 'Manuscript'.

We look forward to receiving your revised manuscript.

Kind regards,

Balew Arega Negatie, Msc,MD

Academic Editor

PLOS ONE

Journal Requirements:

In your cover letter, please note whether your blot/gel image data are in Supporting Information or posted at a public data repository, provide the repository URL if relevant, and provide specific details as to which raw blot/gel images, if any, are not available. Email us at plosone@plos.org if you have any questions

Reviewers' comments:

Reviewer's Responses to Questions

**Comments to the Author**

1. Is the manuscript technically sound, and do the data support the conclusions?

Reviewer #1: Yes

Reviewer #2: Yes

2. Has the statistical analysis been performed appropriately and rigorously? 

Reviewer #1: Yes

Reviewer #2: Yes

3. Have the authors made all data underlying the findings in their manuscript fully available?

Reviewer #1: No

Reviewer #2: Yes

4. Is the manuscript presented in an intelligible fashion and written in standard English?

Reviewer #1: Yes

Reviewer #2: Yes

5. Review Comments to the Author

Reviewer #1: 1. Study population needs clarification. It seems similar with that of source population.

2. Exclusion criteria needs revision. Authors said they excluded patients who has been on antibiotic treatment at the time of data collection but realistically they should have excluded those who are on antibiotic treatment for the last 7 days too.

3. How antibiotics used for antimicrobial susceptibility testing were selected?

4. Where did you classified Intermediate susceptibility during data analysis? (Is it susceptible or resistant?)

5. What controls did you used for PCR?

6. Agarose gel electrophoresis image is not clear (please replace it with clear image)

7. Prevalence, risk factors and drug resistance patterns discussion were not done exhaustively (fewer literature were used for comparison purpose), and reasons given for variations of your findings with other researches seems similar throughout discussion.

8. You mentioned "Even if, the place of defecation, level of contamination of the water source, and sharing of home with cattle were not found associated with the occurrence of STEC O157:H7. However, other researchers reported it.." what do you think the reasons for this? You should explain on discussion section.

9. In Conclusion section you mentioned "...teaching the community.." do you mean health education?

10. Some references are very old (Gannon, 1997; Slutsker, 1998; Olorunshola, 2000; Zhang, 2000; Pruimboom-Brees, 2000; Raji, 2003; Naylor, 2003; Kang, 2004; Thrusfield, 2005; Bryce, 2005; Casburn-Jones, 2004; Cheesbrough, 2006; Brenner, 2005). Please replace them with latest literatures.

Reviewer #2: This is an interesting study reflecting the STEC O157:H7 in the study region and may have good input for scientific people in the area of interest. The authors conducted an extensive study, collected a large number of samples, and carried out scientifically sound bacterial assessment. However, the write up and improving the comments and suggestions is good.

6. PLOS authors have the option to publish the peer review history of their article (what does this mean?). If published, this will include your full peer review and any attached files.

Reviewer #1: **Yes: **Dr. Belayneh Regasa Dadi

Reviewer #2: **Yes: **Birhan Agmas Mitiku

---

## [Author Response · Author response to Decision Letter 0]

14 Aug 2023

August1, 2023

To: Plose one

Editor-In-Chief

From: Tigist Engda (PI and corresponding author)

Submission ID: PONE-D-23-13678

Title: Shiga toxin-producing Escherichia coli O157:H7 among diarrheic patients and their cattle in Amhara National Regional State, Ethiopia

Point by point authors’ response

First, we would like to thank and appreciate the reviewers for their critical and constructive comments. We attempted all the questions and concerns raised by chief reviewers point by point as follows:

 I. JOURNAL REQUIREMENTS:

 Author response: Thank you. We have followed all the PLOS One requirements.

2. Your ethics statement should only appear in the Methods section of your manuscript. 

 Author response: Thank you. We shifted the ethics statement to the Method section as recommended and deleted it from the other section. 

3. PLOS ONE now requires that authors provide the original uncropped and unadjusted images underlying all blot or gel results reported in a submission’s figures or Supporting Information files. 

 Author response: Thank you. We replace the clear image as requested 

II. REVIEWERS COMMENTS

Comments for Reviewer I.

Comment: Study population needs clarification. It seems similar with that of source population.

Authors response: Thank you! We have noted and explained as the source populatio Date:n is all diarrheic patients who had visited the respective Comprehensive Specialized Hospitals for medical services and their cattle. However; Those diarrheic patients who are volunteers for the study and fulfilled the requirements to be the study participants (giving informed consent and the required amount of stool samples for laboratory analysis) are the study population. Page5, line 102-105.

 Comment: Exclusion criteria needs revision. The authors said they excluded patients who has been on antibiotic treatment at the time of data collection but realistically they should have excluded those who are on antibiotic treatment for the last 7 days too.

 Authors response: Thank you! Yes of course; those who had taken antibiotics during the past 7 days of data collection were excluded but might be cut during the editing of the document therefore we are now corrected. Page5, line112-114.

 Comment: How antibiotics used for antimicrobial susceptibility testing were selected?

 Authors response: Thank you! It is a good comment, it is selected based on the Clinical and Laboratory Standards Institute (CLSI, 2022) guideline. All the selected drugs were effective for diarrheal diseases caused by E. coli. However, we include norfloxacin which can be used as an effective drug for the treatment of diarrheagenic E. coli (Msolo Luyanda; et al; 2020). In other selected drugs, there is at least one drug frequently prescribed to treat diarrheal diseases caused by bacteria in their class, so we are trying to check the resistance rate of drugs in each class of antimicrobials. (Ref: “Msolo L, Iweriebor B C, Okoh A I. Antimicrobial Resistance Profiles of Diarrheagenic E. coli (DEC) and Salmonella Species Recovered from Diarrheal Patients in Selected Rural Communities of the Amathole District Municipality, Eastern Cape Province, South Africa. Dovepress journal Infection and Drug Resistance, 2020:13 4615–4626”).

 Comment: Where did you classify Intermediate susceptibility during data analysis? (Is it susceptible or resistant?)

 Authors response: Thank you for the concern. Intermediate susceptibility was small in number and included as sensitive. 

 Comment: What controls did you use for PCR?

 Authors response: Thank you! We have noted and corrected it as ‘‘A negative control was performed by adding 10 µl of sterile deionized water; a positive control was performed by adding 1.25µl of known DNA sample (STEC O157:H7; ATCC 43895)’’. Page 10, lines 212-214

Comment: The agarose gel electrophoresis image is not clear (please replace it with a clear image).

 Authors response: Thank you! for the comment, it has been replaced by a clear image.

Comment: The prevalence, risk factors, and drug resistance patterns discussion was not done exhaustively (fewer literature were used for comparison purposes), and the reasons given for variations of your findings with other research seems similar throughout the discussion.

 Authors response: Thank you! Even though there are not much more studies done in our country for comparison. We are now revised.

Comment: You mentioned "Even if, the place of defecation, level of contamination of the water source, and sharing of home with cattle were not found associated with the occurrence of STEC O157:H7. However, other researchers reported it." what do you think are the reasons for this? You should explain in the discussion section.

 Author response: Thank you! for your logical question. First of all, when we see the socio-demographical characteristics of the study population, only 25.7% of STEC O157: H7 positive patients shared their home with cattle, and the remaining (74%) do not. From this data, we cannot expect an association. However, in studies were higher results of sharing a home with cattle, there may be an association and the reason can work. Similarly, in surface water sources and open field defecation was detected in 3.9% and 22.7% of STEC O157:H7 positive respectively, and has no association. This may be due to less positive isolate. 

9. Comment: In the Conclusion section you mentioned "...teaching the community." do you mean health education? 

 Author response: Yes. It means health education. 

 Comment: Some references are very old (Gannon, 1997; Slutsker, 1998; Olorunshola, 2000; Zhang, 2000; Pruimboom-Brees, 2000; Raji, 2003; Naylor, 2003; Kang, 2004; Thrusfield, 2005; Bryce, 2005; Casburn-Jones, 2004; Cheesbrough, 2006; Brenner, 2005) and MacFaddin JF,2000. Please replace them with the latest literature.

 Author response: Thank you. The comment is accepted and replaced by other references.

Comments for Reviewer II.

This is an interesting study reflecting the STEC O157:H7 in the study region and may have good input for scientific people in the area of interest. The authors conducted an extensive study, collected a large number of samples, and carried out scientifically sound bacterial assessments. However, the write-up and improving the following comments and suggestions is good. 

1. Comment: Did the author clarify: Page 2 line number 29…………………Ethiopia from December/2020 to May/2022. Page 4 line 89……………from December 2020 to June 2022. Which one was the exact study period? 

 Authors response: Thank you for the comment. we have noted and the correct one is December 2020 to June 2022. So, it is corrected in the text as’’ December 2020 to June 2022’’.

2. Comment: Line 89: study was carried out……. from December/2020 to May/2022. A total of 1,149 Diarrheic stool samples and 229 cattle Recto-anal mucosal swab samples were collected.” The samples were collected over one year, meaning at different seasons of the year. Did you consider the effects of season on the disease level?

thor’s response: Yes, they are different since around half of human cases are sporadic with a seasonal pattern favoring spring and summer. Also, in this study, 49.2% and 25.8% of STEC O157:H7 isolate was detected in the summer and spring seasons respectively.

 3. Comment: The description of the sampling strategy is not clear. Page 6, line 116 Authors wrote that …………… The study participants were selected using simple random sampling techniques. How could be? Because the study is an institution based; that means diarrheic patients is not known their names before visiting the hospital. Or may not wait for you after they gain service, they leave the hospital except admit ones. So, how can list their name and lottery then sample? Can Authors clarify? I think you mean systematic random sampling???? 

 Author’s response: Yes, the comment is well accepted and corrected as” A simple random sampling technique was used to select Comprehensive Specialized Hospitals. However, the study participants were selected using systematic random sampling techniques”. Page6, line 122and 121.

 4. Comment: Page 6 line numbers 118-120; Twelve-month diarrheic cases were taken from the patient registration book in order to project t2e probable case flow in each hospital per day. Then the study subjects were recruited consecutively until the desired sample was reached. Is not clear; Can Authors elaborate in figures/numbers to clarify the readers?

 Authors response: Thank you! We accepted the comment and corrected it on page 6, lines 122-136.

 5. Comment: In the case of cattle, convenient sampling techniques were used. All cattle found in STEC O157:H7-positive houses were included in the study and were found 229. Not clear; do you mean convenient sampling? Or census/all cattle sample? 

Since hospital service users come from a distance even from politically unstable areas; so, how do you trace back to their cattle to include all cattle in positive houses? E.g. Beshangule gumize, Jawi to Felghiwote hospital, Afar region to Dessie hospital etc…

Authors response: Yes. the comments are well accepted and corrected as ‘’all cattle found in the STEC O157:H7 positive houses were included in the study.’’ Page 6, lines 132 to 134. Of course, geographical distribution (highly remote areas), and politically unstable areas are challenges to achieved the plan of sample collection. However, in our study, the cattle used for sample collection are around the selected hospital. this means; it is not as far or remote. In addition; the specific house was found with the help of animal science personnel at the site and registered STEC O157:H7 positive diarrheic patient profile (Name, phone no., specific name of the site (kebele name, ‘Gote’ name)).

 6. Comment: A scholar suggests???? E. coli is may originated from human, or other animal feces not only cattle.” I’m afraid how to handle another source of infection other than cattle. E.g. sheep, dogs, other non-diarrheic humans etc… 

 Authors response: Thank you! of course, animals, other than cattle and asymptomatic humans (non-diarrheic humans) can be the source of STEC O157:H7. So, to manage the distribution of STEC O157:H7, we have to prevent the root of transmission by eating cooked and non-contaminated food, having proper contact with animals and also human

7. Comment: Page 12 line 257…………..categorical variables, univariable and multivariate analysis with a 95% confidence interval (CI)………………change by categorical variables, bi-variable and multivariable logistic regression analysis with 95% confidence interval(CI)……………………. Because to make uniformity, and multivariate analysis and multivariable analysis are quite different.

 Author’s Response: Yes, multivariate analysis and multivariable analysis are completely different and the comment is well taken and corrected in the main document.

8. Comment: Page 29 your conclusion part line # 421-422 Shiga toxin-producing E. coli O157:H7 isolates developed drug resistance to most antimicrobials tested. Giving antibiotic treatment in E.coli infection of humans and animal causes worse of the disease thus not recommended; despite this, where is the drug resistance come from?

 Authors response: It is a good comment. In most laboratory setups of the country, the specific bacterial organism that causes diarrhea has not been identified rather is reported as bacteria if there are pus cells in the field. Based on this simple report, the prescriber just prescribes any of the antibiotics that are available in health institutions. Therefore; the patient may be repeatedly exposed to these antibiotics and due to lack of knowledge, most patients did not have the habit of taking medicine completely which altogether contributes to the development of resistance. Another possibility is that mutant bacteria may be directly reached to the patient and consequently cattle can consume these mutant bacteria or vice versa.

 Comment: In addition, u tested the number of antibiotics that are ban and not used in animals? E.g., Chloramphenicol; what is the reason for the discrepancy?

 Authors response: Thank you! for the comment. Even though sensitivity tests had been done for banned antibiotics in cattle, it still gives a clue that the cattle can consume the already resistant bacteria for the drug. It also gives information on the transmission of STEC O157:H7 from humans, the environment, plants, soil, and water to animals.

 We thank all the reviewers for giving comments and supportive suggestions in order to improve the paper!

Please note all the corrections made were labeled in red color in the main text. 

With best regards

 Tigist Engda (PI and corresponding author)

---

## [Decision Letter · Decision Letter 1]

1 Oct 2023

PONE-D-23-13678R1Shiga toxin-producing Escherichia coli O157:H7 among diarrheic patients and their cattle in Amhara National Regional State, EthiopiaPLOS ONE

Dear Dr. Engda,

Thank you for submitting your manuscript to PLOS ONE. After careful consideration, we feel that it has merit but does not fully meet PLOS ONE’s publication criteria as it currently stands. Therefore, we invite you to submit a revised version of the manuscript that addresses the points raised during the review process.

We look forward to receiving your revised manuscript.

Kind regards,

Balew Arega Negatie, Msc,MD

Academic Editor

PLOS ONE

Additional Editor Comments:

Dear author

Your manuscript needs further and extensive work before we consider it for publication. I attached my comments and suggestions as highlighted in the main manuscript.

Reviewers' comments:

Reviewer's Responses to Questions

**Comments to the Author**

1. If the authors have adequately addressed your comments raised in a previous round of review and you feel that this manuscript is now acceptable for publication, you may indicate that here to bypass the “Comments to the Author” section, enter your conflict of interest statement in the “Confidential to Editor” section, and submit your "Accept" recommendation.

Reviewer #1: All comments have been addressed

2. Is the manuscript technically sound, and do the data support the conclusions?

Reviewer #1: Yes

3. Has the statistical analysis been performed appropriately and rigorously? 

Reviewer #1: Yes

4. Have the authors made all data underlying the findings in their manuscript fully available?

Reviewer #1: No

5. Is the manuscript presented in an intelligible fashion and written in standard English?

Reviewer #1: Yes

6. Review Comments to the Author

Reviewer #1: (No Response)

7. PLOS authors have the option to publish the peer review history of their article (what does this mean?). If published, this will include your full peer review and any attached files.

Reviewer #1: **Yes: **Dr. Belayneh Regasa Dadi

---

## [Author Response · Author response to Decision Letter 1]

29 Oct 2023

Thank you , 

we have been attached the academic editor( Balew Arega Negatie, Msc,MD) and reviewer (Dr. Belayneh Regasa Dadi) comments with specific response one by one. 

Date: October 18, 2023

To: Plose one

Editor-In-Chief

From: Tigist Engda (PI and corresponding author)

Submission ID: PONE-D-23-13678

Title: Shiga toxin-producing Escherichia coli O157:H7 among diarrheic patients and their cattle in Amhara National Regional State, Ethiopia

Point by point authors’ response

First, we would like to thank and appreciate the reviewers for their critical and constructive comments. We attempted all the questions and concerns raised by academic editors and reviewers. We used red font color for changes. 

 I. ACADEMIC EDITOR’S COMMENTS:

1. Comment: Please include the following items when submitting your revised manuscript:

• An unmarked version of your revised paper without tracked changes. You should upload this as a separate file labeled 'Manuscript'

Author response: Thank you! We have included all items; the rebuttal letter (Response to Reviewers), a marked-up copy of the manuscript with file name “Revised Manuscript with Track changes” and unmarked version of the original file with file name “Manuscript” are attached and you are kindly requested to upload these files. 

2. Comment: Your manuscript needs further and extensive work before we consider it for publication. I attached my comments and suggestions as highlighted in the main manuscript

Authors response: Thank you! We have noted and corrected it point by point in the manuscript and manuscript with track changes. 

2.1. Comment: Line 37: make it ‘‘under five children’’.

 Author response: Thank you! the comment is well accepted and corrected it on page 2, line 37. 

2.2. Comment: Line 38 &39, patients with bloody diarrhea 39 (AOR:4.09, 95%CI:2.09-8.00). This is the manifestation of the disease how does it become a risk factor?

 Author response: Thank you! for the comment and corrected it on page 2, line 39. But what we want to show is the distribution of the organism with patient’s stool appearance and actually it is the problem of write up and were not the place here and removed. 

2.3. Comment: Line 57: Delete the word ‘to’

 Author response: Thank you! we have noted and corrected it on page 3, line 57

2.4. Comment: Line 59: The word ‘which’ change by ‘that’

 Author response: Thank you! for the comment and corrected it on page 3, lines 59

2.5. Comment: line 105: The word ‘Those’ change by ‘These’

 Author response: Thank you! we have noted and corrected it on page 5, line 104.

2.6. Comment: line115: The word ‘that’ change by ‘who’.

 Author response: Thank you! for the comment and corrected it on page 6, line 111.

2.7. Comment: line 137: add the word ‘the’.

 Author response: Thank you. We accepted the comment and corrected it on page 7, line132

2.8. Comment: line 155: What is this?

 Author response: Thank you! we have noted that the number 2 was deleted in mistake during editing and corrected it on page 7, line 150 as “2gm”.

2.9. Comment: line 199: Correct this.

 Author response: Thank you! for reading critically it was the bracket that indicates the reference number that we take during first draft writing and now it is removed and corrected on page 9, line 194 as “analysis”.

2.10. Comment: line 243: The word ‘became’ change to ‘were’.

 Author response: Thank you! We have noted and corrected it on page 11, line 235.

2.11. Comment: line 246: The word ‘placing’ change to ‘placed’.

 Author response: Thank you! for the comment and corrected it on page11, line 238.

2.12. Comment: line 249: Put clear definition of MDR.it means to be resistance to all 

 mentioned?

 Author response: Thank you! for the comment. No, it does not mean to be resistance to all the selected antibiotics. However, Multiple Drug Resistance is defined as “resistance to at least three antibiotic classes”. It is defined in page 12, line 241. These classes of antibiotics are listed on page 12, line 242-246 

2.13. Comment: line 265: Delete the word ‘one’.

 Author response: Thank you! we have rewrite it as “a week prior to data collection” on page 12, line 257.

2.14. Comment: line 270: Delete the word ‘own’.

 Author response: Thank you! for the comment and we have corrected it on page 12, line 262.

2.15. Comment: line 281:is it colonization or infection? Specially among diarrheic patients.

 Author response: Thank you! we have noted and corrected it on page 13, line 272

 as “infection”.

2. 16. Comment: line 294: The word ‘explained’ change to ‘explaining’.

 Author response: Thank you! for the comment and we corrected it on page 13, line 284.

2.17. Comment: line 298: ‘in order to’ change to’ to’

 Author response: Thank you.! we have noted and corrected it on page 14, line 288.

2.18. Comment: line 312: This idea (…However, from 187 diarrheic patients, only 53 study participants were positive for STEC O157:H7 which had a total of 229 cattle with 124 cows, 46 oxen, and 59 calves) good to take it to the prevalence subtitle below.

 Author response: Thank you! we have noted and shifted it to under the subtitle of Prevalence of shiga toxin-producing Escherichia coli O157:H7 on page 16, line 311-314.

2.19. Comment: line 318: The word ‘sample’ change to ‘samples.’ 

 Author response: Thank you! for the comment and we have corrected it on page 16, line 303.

2.20. Comment: line 320-323: These sentences are repeated several times. Please summarize to

 avoid repetition and add CI.

 Author response: Thank you. We have noted and rewrite it on page 16, line 303-306. as “of the total stool samples, 128 stool samples were positive for STEC O157:H7 with rfb O157 and flic H7 genes in PCR with agarose gel Electrophoresis tested (Table1). As a result, the overall prevalence was 11.1% (95% CI: 0.09-0.13)”.

2.21. Comment: line 328: Please include the details as a supplementary, which cattle, how many cattle from single index diarrheic patient. 

 Author response: Thank you! for the comment and detailed explanation about cattle have discussed on page 16, line from 312 to 317.

2.22. Comment: line 330-334: re-summarize the sentence “there were 33 isolates which has rfb O157 genes. These rfb O157 positive isolates were tested for the presence of flic H7 gene. All the 33 isolates in cattle had both rfb (157 and flic H7 as a result the overall prevalence of STEC O157:H7 in cattle was 33/229 (14.4%; 95%CI:0.39-0.52). 

 Author response: Thank you! the sentence is re-summarized on page 16, line 315-318. 

2.23. Comment: line 339: insert ‘a’ in front of higher.

 Author response: Thank you! we have noted and corrected in page 18, line 326.

 as ‘a higher’.

2.24. Comment: line 340: rewrite ‘the age group of under 5’ to ‘among under 5 children’.

 Author response: Thank you! for the comment and we have rewrite it on page 18, line 327.

2.25. Comment: line 355: Please remove variables that did not full fill your eligibility criteria (p<0.25), because has no importance here.

 Author response: Thank you! We accepted the comment and corrected it on page 19-21. Table 4. We removed the variables that did not full fill the eligibility criteria.

2.26. Comment: in table 4, (stool appearance variable): This is difficult to interpret it is an established fact that E.coli O157:H7 is invasive and causes bloody diarrhea. Manifestation can not be the risk factor for the given disease please avoid. 

 Author response: Thank you! we accepted the comment and removed stool appearance in the analysis from Table 4. However; it was analyzed in percent to know the distribution of the organism with patient’s stool appearance and explained under the prevalence subtitle.

2.27. Comment: line 377: Table6 is only for human not from cattle

 Author response: Thank you! yes, table 6 is only for human and corrected it on page 23, line 359-362. Of-course, it is editing problem. However, the description for cattle has its place page 24, line 362 and Table 7. 

2.28. Comment: line 383: This table is hard to understand? It is not simple, it is complex, for the table 6. Please write short summary in sentences and take them to supplementary, be sure also that, these tables give more information not included in the table 5.

 Author response: Thank you! for the comment and incorporated brief summary on page 23, line 359-362. Table 5 indicates the susceptibility profile for STEC O157:H7 isolates from diarrheic patients and their cattle however Tabel 6 indicates antibiogram and there is no repetition. 

2.29. Comment: in table 6 of the second row: what is this? title is Antibiotics disk.

 Author response: Thank you! for the comment. It is not title but it is a total number of isolates in diarrheic patients. However, it is now removed because it is already written at the bottom of the table 6.

2.30. Comment: in the table 6, from the third to last rows: What this numbers means? Similar to others?

 Author response: Thank you! we have noted and discussed it on page 24, line 368-369 as “Number in bracket: the number of isolates which showed resistance to the listed antibiotic classes’’ (page 24, line 368-369). For example, table 6, row 3, “TC (5)”, this means five isolates showed resistance for only tetracycline. Row 4 “AMC, TC (5)” this means 5 isolates showed resistance to amoxicillin/clavulanic acid and tetracycline. Since these two drugs are from different antibiotic classes therefore their antibiogram categorization is R2.

2.31. Comment: line 390: Table7 is only for human not from cattle

 Author response: Thank you! Yes, table 7 is only for cattle and corrected it on page 24, line 372. Of-course, it is editing problem. 

2.32. Comment: line 390: This table is hard to understand? It is not simple, it is complex, for the table 7. Please write short summary in sentences and take them to supplementary, be sure also that, these tables give more information not included in the table 5.

 Author response: Thank you! for the comment and incorporated brief summary on page 24, line 370-372. Table 5 indicates the susceptibility profile for STEC O157:H7 isolates from diarrheic patients and their cattle however Table 7 indicates antibiogram and there is no repetition.

2.33. Comment: in table 7 second row what is this? title is Antibiotics disk. row

 Author response: Thank you! for the comment. It is not title but it is a total number of isolates in cattle. However, it is now removed because it is already written at the bottom of table 7.

2.34. Comment: in table 7 from third to last rows: What this numbers means? Similar to others?

 Author response: Thank you! we have noted and give an explanation on page 26, line 378-379 as “Number in bracket: the number of isolates which showed resistance to the listed antibiotic classes’’(page26,line 378-379). For example, table 7, row 3 “SXT (1)” this means one isolate is resistance to only Sulfamethoxazole/Trimethoprim. Row 4 “AMC, TC (1)” this means 1 isolate showed resistance to amoxicillin/clavulanic acid and tetracycline. Since these two drugs are from different antibiotic classes therefore from the antibiogram, they are categorized as R2. 

2.35. Comment: line 398. Please in the first paragraph include the short summery of your main findings.

 Author response: Thank you! for the comment. We have included summery of our main findings on page 26, line 381-388.

2.36. Comment: line 407-410. This is a very general explanation rather than comparing numbers. Interpret your findings apply throughout your discussion 

 Author response: thank you! we have noted and corrected though out the discussion on page 26-28.

2.37. Comment: line 420. Please remove the AOR, CI and P-value from your discussion part. Do for the rest.

 Author response: Thank you! for the comment. We have seen and removed AOR, CI and P-value in discussion part.

2.38. Comment: line 436: change the word ‘in applying’ by ‘to apply’.

 Author response: Thank you! for the comment and we have deleted the word ‘in applying’ and replace by ‘to apply’ on page 27, line 419.

2.39. Comment: line 452 and 553: remove the word since and even. 

 Author response: Thank you! for the comment and we have deleted the word “since” before animal on page 28, line 434 and “even” before some on page 28, line 435 respectively.

2.40. Comment: Line 469. Please include the limitation of the study

 Author response: Thank you! for the comment. We have incorporated the limitation of our study on page 29, line 452-453

2.41. Comment: replace the word “for” by “to”, “for” by “to”, “of” by “to” and remove the word “on” in page29, line 463, 463, 464, and 463 respectively.

 Author response: Thank you! for the comments. we accept the comments and corrected on page 29, line 463.

In general, we have tried to edit the whole body of the paper. 

We thank all the academic editors and reviewers for giving comments and supportive suggestions in order to improve the paper!

Please note all the corrections 

With best regards!

Tigist Engda (PI and corresponding author)

---

## [Editor Report · Decision Letter 2]

10 Nov 2023

PONE-D-23-13678R2Shiga toxin-producing Escherichia coli O157:H7 among diarrheic patients and their cattle in Amhara National Regional State, EthiopiaPLOS ONE

Dear Dr. Engda,

Thank you for submitting your manuscript to PLOS ONE. After careful consideration, we feel that it needs  minor modification  to fully meet PLOS ONE’s publication criteria as it currently stands. Therefore, we invite you to submit a revised version of the manuscript that addresses the points raised during the review process.

ACADEMIC EDITOR: Please insert comments here and delete this placeholder text when finished. Be sure to:Indicate which changes you require for acceptance versus which changes you recommendThank you addressing all comments and suggestions in the previous version. Here , I attached some minor comments as track change in the main manuscriptPlease submit your revised manuscript by Dec 25 2023 11:59PM. If you will need more time than this to complete your revisions, please reply to this message or contact the journal office at plosone@plos.org. Please include the following items when submitting your revised manuscript:A rebuttal letter that responds to each point raised by the academic editor and reviewer(s). You should upload this letter as a separate file labeled 'Response to Reviewers'.A marked-up copy of your manuscript that highlights changes made to the original version. You should upload this as a separate file labeled 'Revised Manuscript with Track Changes'.An unmarked version of your revised paper without tracked changes. You should upload this as a separate file labeled 'Manuscript'.We look forward to receiving your revised manuscript.

Kind regards,

Balew Arega Negatie, Msc,MD

Academic Editor

PLOS ONE
---

## [Author Response · Author response to Decision Letter 2]

13 Nov 2023

Date: November 12, 2023

To: Plose one

Editor-In-Chief

From: Tigist Engda (PI and corresponding author)

Submission ID: PONE-D-23-13678 R2

Title: Shiga toxin-producing Escherichia coli O157:H7 among diarrheic patients and their cattle in Amhara National Regional State, Ethiopia

Point by point authors’ response

First, we would like to thank and appreciate the reviewers for their critical and constructive comments. We attempted all the questions and concerns raised by academic editors.

 ACADEMIC EDITOR’S COMMENTS:

1. Requirement at time of submitting revised manuscript:

 Author response: Thank you! We have included all items; the rebuttal letter (Response to Reviewers), a marked-up copy of the manuscript with file name “Revised Manuscript -R4 with Track changes” and unmarked version of the original file with file name “Manuscript” are attached and you are kindly requested to upload these files. 

II. Comment: Please insert comments here and delete this placeholder text when finished.

 Be sure to:

1. Indicate which changes you require for acceptance versus which changes you recommend 

 Authors response: Thank you. We consider it.

2. Thank you for addressing all comments and suggestions in the previous version. Here, I attached some minor comments as track changes in the main manuscript.

Author response: Thank you. We have noted and corrected it point by point in the manuscript and manuscript-R4 with track changes. 

 2.1. Comment: Line 48: the word ‘continues’ change by ‘continued’

 Author response: Thank you for the comment and corrected it on page3, line 48.

 2.2. Comment: Line 173: The word ‘which’ change by ‘that’

 Author response: Thank you. We have noted and corrected it on page 8, line 171.

2.3. Comment: Line 206: The word ‘encoded’ change by ‘encoding’

 Author response: Thank you. We have noted and corrected it on page 10, line 204.

2.4. Comment: Line 223: add the letter ‘a’ before the word gel.

 Author response: Thank you for the comment and corrected it on page 10, line 220.

2.5. Comment: Line 243: Delete the word ‘by’.

 Author response: Thank you for the comment and corrected it on page11, line 238.

2.6. Comment: Line 262: The word ‘prior to’ change by ‘before’ and add the word ‘the’ before training. 

 Author response: Thank you. We have noted and corrected it on page 12, line 257.

2.7. Comment: Line274-276: sentences state in this line and the sentences on line 273-274 are opposites; please remove this one. You only variables with p<0.2 in bivariate to multivariable regression.

 Author response: Thank you. We accepted the comment and corrected it on page 13. Line 267-270.

2.8. Comment: Line 281: edit the word ‘lemeshow’ by ‘Lemeshow’

 Author response: Thank you for the comment and corrected it on page 13, line 274.

2.9. Comment: Line 290: Delete the word on

 Author response: Thank you for the comment and corrected it on page13, line 282.

2.10. Comment: Line 293: Delete H7

 Author response: Thank you for the comment. We were communicated with those who were STEC O157:H7 positive patients not all STEC O157 positive patients. Basically, STEC O157:H7 is one group of the STEC O157. So, it should be stated as STEC O157:H7. Because our aim is to find out only STEC O157:H7.

2.11. Comment: Line 319: The word ‘from’ change by the word ‘of’

 Author response: Thank you. we are corrected on page 16, line 313.

2.12. Comment: Line 322: the word ‘swabs’ change to ‘swab’

 Author response: Thank you. we are corrected on page16, line 315.

2.13. Comment: Line 334 and 335: the word ‘were’ and resident change by ‘was’ and ‘residents’ respectively

 Author response: Thank you. We are corrected on page 18. Line 327 and 328 respectively.

2.14. Comment: Line 347, table 4: What is the value of AOR for drinking water source and drinking water treatment?

 Author response: Thank you. We are now incorporated the AOR value of drinking water source and drinking water treatment on page 20, line 340 in table 4.

2.15. Comment: Line 347, table 4: why do you include this variable in Multivariable regression because the p value is greater than your selection criteria, P<0.2 in History of hand washing habit independent variable.

 Author response: Thank you for the comment. The P-value of after eating, one of the categories under History of hand washing habit variable, is (0.00) which is less than 0.2. and needs to calculate Multivariable regression to know it’s association. Similarly drinking water treatment variable too.

2. 16. Comment: Line 377: change the word ‘which’ by ‘that’.

 Author response: Thank you. We are corrected on page 24, line 369.

2.17. Comment: Line 380: change the word ‘antibiotics’ to ‘antibiotic’.

 Author response: Thank you and corrected on page 24, line 372

2.18. Comment: Line 393: delete the word ‘and” before the highest.

 Author response: Thank you and corrected on page 26, line 385.

2.19. Comment: Line 393: add hyphen b/n under and five.

 Author response: Thank you and corrected on page 26, line 385.

2.20. Comment: line 394: add the word ‘of’ after 11.7% and after 6.1%.

 Author response: Thank you and corrected on page 26, line 386.

2.21. Comment: Line 396: change the word ‘were’ by ‘was.

 Author response: Thank you and corrected on page 26, line 388.

2.22. Comment: Line 407: Change the word ‘traditionally’ to ‘traditional’.

 Author response: Thank you and corrected on page 26, line 398.

2.23. Comment: Line 413: change the word ‘to’ by ‘in’.

 Author response: Thank you and corrected on page 26, line 404.

2.24. Comment: line 414: add hyphen after the word ‘under’, change the word ‘was’ by ‘were’ and ‘to’ by ‘for’.

 Author response: Thank you and corrected on page 27, line 405. 

2.25. Comment: line 415 and 416: add ‘s’ to the word system and practice and the before time.

 Author response: thank you and corrected on page 27, line 406.

2.26. Comment: Line 419: the word had been changed by were.

 Author response: Thank you and corrected on page 27, line 409.

2.27. Comment: Line 421: change the word ‘are’ by ‘is’.

 Author response: Thank you and corrected on page 27, line 412.

2.28. Comment: Line 426: change the word was by were.

 Author response: Thank you and corrected on page 27, line 417.

2.29. Comment: Line 434: add the word ‘and’ before Mexico and Line 435: add the word ‘and’ before the type of samples.

 Author response: Thank you and corrected on page 28, and line 425 and 426.

2.30. Comment: Line 451: Delete the word ‘in’ before different.

 Author response: Thank you and corrected on page 28, line 442.

2.31. Comment: Line 460: add word ‘and’ before means, ‘comma’ after transmission and ‘a’ before prevention.

 Author response: Thank you. we have incorporated the word “and” before the word “means” added comma after transmission and “a” before prevention. Page 29, line 451.

2.32. Comment: Line 473: add ‘ing’ to the word ‘teach’

 Author response: Thank you and corrected on page 29, line 463.

2.33. Comment: Line 475: add the word ‘and’ before the word after

 Author response: Thank you and corrected on page 29, line 465

2.34. Comment: Line477: change the word ‘with respect to’ by ‘concerning’.

 Author response: Thank you and changed the word on page 29, line 466.

NB: We appreciate the academic editors for critically checked the manuscript and giving us valuable comments and of course we have tried to attempt all the comments. 

With best regards!

Tigist Engda (PI and corresponding author)

---

## [Editor Report · Decision Letter 3]

20 Nov 2023

Shiga toxin-producing Escherichia coli O157:H7 among diarrheic patients and their cattle in Amhara National Regional State, Ethiopia

PONE-D-23-13678R3

Dear Dr. Tigist

We’re pleased to inform you that your manuscript has been judged scientifically suitable for publication and will be formally accepted for publication once it meets all outstanding technical requirements.

Kind regards,

Balew Arega Negatie, Msc,MD

Academic Editor

PLOS ONE
---

## [Editor Report · Acceptance letter]

12 Dec 2023

PONE-D-23-13678R3 

PLOS ONE

Dear Dr. Engda, 

I'm pleased to inform you that your manuscript has been deemed suitable for publication in PLOS ONE. Congratulations! Your manuscript is now being handed over to our production team.

Kind regards, 

on behalf of

Dr. Balew Arega Negatie 

Academic Editor

PLOS ONE